# Utilisation of a Suite of Screening Tools to Determine Adverse Healthcare Outcomes in an Older Frail Population Admitted to a Community Virtual Ward

**DOI:** 10.3390/ijerph18115601

**Published:** 2021-05-24

**Authors:** Clare Lewis, Rónán O’Caoimh, Declan Patton, Tom O’Connor, Zena Moore, Linda E. Nugent

**Affiliations:** 1School of Nursing and Midwifery, Royal College of Surgeons Ireland, 123 St Stephen’s Green, Saint Peter’s, D02 YN77 Dublin, Ireland; declanpatton@rcsi.ie (D.P.); tomoconnor@rcsi.ie (T.O.); zmoore@rcsi.ie (Z.M.); lindanugent@rcsi.ie (L.E.N.); 2Clinical Sciences Institute, National University of Ireland Galway, Costello Road, H91 TK33 Galway City, Ireland; rocaoimh@hotmail.com; 3Department of Geriatric Medicine, Mercy University Hospital, Grenville Place, T12 WE28 Cork City, Ireland

**Keywords:** risk screening, clinical health states, older persons, community, virtual wards

## Abstract

Risk stratification to assess healthcare outcomes among older people is challenging due to the interplay of multiple syndromes and conditions. Different short risk-screening tools can assist but the most useful instruments to predict responses and outcomes following interventions are unknown. We examined the relationship between a suite of screening tools and risk of adverse outcomes (pre-determined clinical ‘decline’ i.e., becoming ‘unstable’ or ‘deteriorating’ at 60–90 days, and institutionalisation, hospitalisation and death at 120 days), among community dwellers (n = 88) after admission to a single-centre, Irish, Community Virtual Ward (CVW). The mean age of patients was 82.8 (±6.4) years. Most were severely frail, with mean Clinical Frailty Scale (CFS) scores of 6.8 ± 1.33. Several instruments were useful in predicting ‘decline’ and other healthcare outcomes. After adjustment for age and gender, higher frailty levels, odds ratio (OR) 3.29, (*p* = 0.002), impaired cognition (Mini Mental State Examination; OR 4.23, *p* < 0.001), lower mobility (modified FIM) (OR 3.08, *p* < 0.001) and reduced functional level (Barthel Index; OR 6.39, *p* < 0.001) were significantly associated with clinical ‘decline’ at 90 days. Prolonged (>30 s) TUG times (OR 1.27, *p* = 0.023) and higher CFS scores (OR 2.29, *p* = 0.045) were associated with institutionalisation. Only TUG scores were associated with hospitalisation and only CFS, MMSE and Barthel scores at baseline were associated with mortality. Utilisation of a multidimensional suite of risk-screening tools across a range of domains measuring frailty, mobility and cognition can help predict clinical ‘decline’ for an already frail older population. Their association with other outcomes was less useful. A better understanding of the utility of these instruments in vulnerable populations will provide a framework to inform the impact of interventions and assist in decision-making and anticipatory care planning for older patients in CVW models.

## 1. Introduction

There is a discernible shift in the assessment of older persons care from disease manifestation and trajectories to healthy ageing and enhancement of care through targeted individualized interventions. The World Report on Ageing and Health, conceptualizes the core domains of care as being person-centred, integrated and that enhance activities of daily living with a focus on bridging the gap between achieving the optimal clinical outcomes and supporting day-to-day living that enables people to age and live well [1]. These include underlying determinants of health such as nutrition, access to food and functioning in a safe and healthy environment [1]. An enabler of this is the utilization of screening tools that assess the level of dependency, the environment in which care is delivered and early identification and intervention [1]. This requires a multifactorial assessment framework to assess the complexity of care needs. 

Assessing complexity and determining care dependency levels is becoming increasing challenging in older populations [2]. This is in part due to growing levels of multi-morbidity [3,4] and a high prevalence of frailty in older populations [5]. This has resulted in the need to provide increasingly complex care in the community [6]. Older persons are nevertheless a heterogeneous population with varying health and social care requirements. This increases the need to develop risk-predication models that are multidimensional in assessing and determining responses with a focus on proactive rather than re-active approaches to care to prevent further disability [6]. While risk-prediction is important to determine these needs, there are few instruments or models available to accurately inform decision-making and support healthcare professionals to manage this complexity [7].

Community Virtual Wards (CVW) have evolved and developed to provide a model of care that can identify healthcare outcomes earlier and the resources required. A CVW provides an environment to support risk stratification and deliver case management approaches to care [6,7]. Case management facilitates team-based approaches, addresses unmet need and improves care co-ordination and well-being [6,7,8,9,10]. A CVW is defined as a multi-disciplinary group of trained healthcare professionals providing targeted interventions to older populations living in the community [6,7]. This model is targeted towards groups at moderate to higher risk of hospitalisation with complex care needs [6,7]. Care is normally co-ordinated by experienced case managers (a senior nurse) to determine the level of need and services required working with the wider healthcare team in both the acute hospital and community [6,7].

With increasing demand for care models to help assess such complexity and determine healthcare outcomes in the community, understanding how commonly used short screening scores perform in ‘real-life’ clinical practice is important to provide a framework that supports healthcare professionals to determine the likelihood of a positive outcome to care delivery [8]. This study examines the association of several short, risk-prediction instruments with the likelihood of achieving one of three ‘health states’ while admitted to the CVW. These were based on pre-defined health criteria (‘unstable’ or ‘deteriorating’ versus ‘stable’, see Table 1), developed from previous standards for clinical stability in the management of acute medical conditions and measured by the clinical CVW manager (‘Matron’) [9,10,11,12,13,14,15]. Risk of other adverse healthcare outcomes including institutionalisation, hospitalisation and death among older patients while admitted to a CVW were also assessed [9,10,11,12,13,14,15]. 

## 2. Methods and Materials

### 2.1. Study Design

A CVW was developed in North Dublin, Ireland, to support community-dwelling older persons with frailty to reduce the need for institutional care or emergency department (ED) presentations and to limit the number of unplanned hospital admissions [6]. This was achieved using a comprehensive assessment of care needs, informed by a suite of screening tools, and active monitoring to track responses during admission to the CVW. This was followed by targeted interventions [6,7,8,9,10]. Screening tools were scored on admission and at 60 and 90 days to allow a sufficient time period to observe changes. Although previous research on CVWs has employed risk stratification methods using data on co-morbidities, disability and activity in primary and secondary care [14,15,16,17], there is little evidence linking changes in scores over time to determine healthcare outcomes and to predict if patients will reach a level of stability or deteriorate, remain at home or require institutionalisation. This study used a non-experimental correlational design employing a decision-making Markov model to determine domains of risk that influenced transitions through different pre-defined ‘health states’ (Table 1). Data were collected prospectively from a single centre over 90 days after admission to the North Dublin CVW.

### 2.2. Sample

A total of 88 patients were recruited to the study following an initial assessment as part of the usual admission pathway to the CVW. The primary inclusion criteria were age, (≥65 years) and evidence of frailty (stratified following clinical assessment). Further, all patients were community dwellers (living in their own or their caregiver’s home and not in residential care). It was important that patients lived at home as the level of care provided within residential care homes significantly differs with a higher level of monitoring, interventions and nursing input required in comparison to home care [6]. Patients admitted to the CVW were referred from their General Practitioner (GP) and/or Specialist Geriatric Services (SGS) based in the catchment area. All were over 75 years of age and were community dwellers (i.e., living in their own or a relative’s home and not in residential care), with evidence of deterioration in their medical, functional and/or cognitive status during the preceding three months (as judged by their referring physician). 

The North Dublin CVW model operates across three virtual wards that are colour-coded based on level of risk: green (low-risk), amber (moderate-risk) and red (high-risk), see Table 2 [6]. All patients included in the study had experienced an event in the last 30 days and were admitted to the red (high-risk) virtual ward. The level of monitoring was higher in the red CVW with daily visits and types of interventions emulating hospital-at-home approaches to care such as subcutaneous fluids and intravenous antibiotics [6]. Patients were triaged and assessed by the CVW case manager before being transferred to lower virtual ward levels after a period of case management (14–30 days). The pathways of care following discharge were to the GP and the primary care team after approximately 90 days of case management. The characteristics of patients in the CVW have been published elsewhere [6] but are summarised in Table 3 and the Results Section.

### 2.3. Ethics Approval 

This study was conducted in accordance with the Declaration of Helsinki. Ethics approval was obtained from the research ethics committee of the Royal College of Surgeons Ireland on 21 April 2016 (REC1219). Patients provided written informed consent after admission. Assent was obtained by family or a legal representative if required following processes outlined in Ireland’s Assisted Decision Making Capacity Act 2015 [18]. Data were coded on admission to the CVW to protect the confidentiality and anonymity of each patient. 

### 2.4. Data Collection and Outcomes

The study was conducted over a period of 16 months between March 2016 and July 2017 to ensure that all patients involved in the study had completed a CVW admission with a follow-up period for a maximum of 120 days (approximately 4 months) from admission. Descriptive data included demographic details, co-morbidities, number of medications, social history (including signs suggestive of self-neglect, see below) and continence status and the number of hospital admissions, ED presentations and falls in the previous three months (Table 3). The primary outcome of this study was the number achieving stability or declining, defined as the composite of those who remained ‘unstable’ or were ‘deteriorating’. Secondary outcomes included the number of patients institutionalised, hospitalised and dead within 120 days of admission. Institutionalisation was defined as approval to admit to long-term care after presentation and discussion at a local placement forum (includes consultant geriatrician, nursing and health and social care professionals who approve applications to long-term nursing home care). Data were collected from the acute hospital patient case records/charts, GP databases and from the records of the primary care team providing care during the CVW admission. Data were collected at baseline, 60 days and 90 days. This timeframe allowed for the longer period of recovery that is often required in the presence of frailty [19]. 

There were several risk-screening instruments employed and measured as part of a standard assessment to determine care needs. These tools were selected as they are commonly used as part of clinical practice in the community services and SGS in North Dublin (Table 4). Frailty status was stratified using the Rockwood Clinical Frailty Scale (CFS) [20,21]. The CFS is a nine-point scale measuring the level of frailty from robust and vulnerable stages through to mild, moderate, severe, very severe and terminal stages of frailty with scores ≥ 5 indicating frailty [20,21]. Unlike the other measures, it was only recorded at baseline and 60 days. Mobility was measured using the Timed up and Go (TUG) test cut-off of > 13 s, indicating reduced mobility [22], and a modified version of the Functional Independence Measure (FIM). The modified FIM covers 10 categories of functional levels with a cut off > 1 (modified independent) to a maximum of 10 (hoist dependent) [23,24]. A modified version of the Barthel Index (BI) was used to assess basic activities of daily living with scores up to 20 points (independent) (cut-off < 16 indicates low dependency) and a lower score of 5 indicating maximum dependency levels [25]. Pressure ulcer risk was measured using the Walsall pressure ulcer risk tool (cut-off > 3 indicates high risk) [26]. The Malnutrition Universal Screening Tool (MUST) assessed nutritional status with a cut-off of 1 (moderate risk) to ≥2 (high risk) [27,28]. Cognition was measured using the 30-point Mini-Mental State Examination (MMSE), with advanced levels of cognitive impairment defined by MMSE scores in the range of 0–30 points (cut-off score of 26) [29]. Mood was measured using the 15-point Geriatric Depression Scale (GDS) (≥5 suggesting depression) [30]. The Identification of Seniors at Risk (ISAR) tool assessed the overall risk of a hospitalisation (cut-off ≥ 2 signals increased risk of hospitalisation) [31]. 

### 2.5. Measures

Each of the individual risk-screening tools were measured on admission to the CVW (0 days) and at 60 and 90 days (except the CFS, which was not scored at 90 days). These time periods were selected as part of a Markov model to allow each patient time to transition from one state (measured on admission) to another state (at 60 days) that included sufficient recovery time following an acute event or deterioration within a disease pathway (Figure 1).

### 2.6. Data Analysis

Data were analysed using STATA version 14.1 [32] (StataCorp LLC, Texas, USA). Most data were non-normally distributed. Spearman’s (rank correlation coefficient) rho (*r*) measured correlations between risk scores with each clinical health state (i.e., composite outcome of remaining ‘unstable’ or ‘deteriorating’ called ‘clinical decline’ versus ‘stable’ at 60 and 90 days). Logistic regression using a generalised linear model, adjusting for patients age and sex (at each time point) was used to examine whether individual risk scores were associated with each clinical health state and [32] and secondary outcomes: risk of institutionalisation (remaining at home versus being institutionalised), hospitalisation (admission rather than ED attendance) and death during the follow-up period after admission. Odds ratios (OR) and 95% confidence intervals (CI) were calculated for each risk score. 

## 3. Results

The characteristics of the 88 patients included are presented in Table 3 and published in detail elsewhere [6]. In summary, patients had a mean age (±standard deviation) of 82.8 ± 6.4 years and had more severe frailty, with a mean CFS score of 6.8 ± 1.33. All were classed as unstable on admission as per criteria provided in Table 1. Most were female, 65.9%. The average length of stay after admission to the CVW was 123 ± 100 days. At last follow-up (July 2017), 38% (33/88) were institutionalised, 20% (18/88) experienced at least one hospitalisation and 13% (12/88) had died. There was a reduction in the number of ED presentations comparing trends in the three months before and while admitted to the CVW (76% prior vs. 30% after, *p* < 0.001), as well as unplanned hospitalisations (65% prior vs. 20.5% after, *p* < 0.001). The number of individuals requiring assistance of one person to mobilise doubled from admission (n = 9, 10%) to 90 days (n = 18, 20%). There were also increases in the proportion requiring a hoist (system to transfer the person), which more than doubled from admission (n = 5, 5%) to discharge (n = 12, 13%), *p* < 0.001.

### 3.1. Relationship between Instruments and Clinical Stability or Decline 

Risk scores were compared to health states at 60 and 90 days after admission to the CVW to examine the significance of both correlations and associations with health states (composite of remaining ‘unstable’ or ‘deteriorating’ versus becoming ‘stable’) for the total sample (n = 88) over time (Table 5 and Table 6).

#### 3.1.1. Correlations

The ISAR was moderately significantly correlated with clinical ‘decline’ (composite outcome) at 60 (*r* = 0.44, *p* < 0.001) and 90 days (*r* = 0.45, *p* < 0.001). Lower BI scores (11–15) at 60 days also showed a significant moderate correlation with decline at 60 (*r* = 0.54, *p* < 0.001) and 90 days, (*r* = 0.60, *p* < 0.001). Lower Rockwood CFS scores (moderate level 6) were likewise correlated with ‘decline’ at 60 days (*r* = 0.57, *p* < 0.001), although this weakened by 90 days (*r* = 0.44, *p* < 0.001). The MUST had only a small-moderate correlation (*r* = 0.22, *p* = 0.039) with ‘decline’ at 60 and at 90 days (*r* = 0.32, *p* = 0.002). Scores on the Walsall, indicating a higher risk of pressure ulcer development, had moderate significant correlation with decline at 60 days (*r* = 0.59, *p* < 0.001), increasing at 90 days (*r* = 0.68, *p* < 0.001). The level of mobility (less independence) measured on admission using the FIM was also significantly and moderately correlated with a ‘decline’ at 60 days (*r* = 0.56, *p* = 0.01) and remained in this state at 90 days (*r* = 0.58, *p* < 0.001). The MMSE showed a weak correlation with ‘decline’ at 60 days (*r* = 0.26), which became moderate over time at 90 days (*r* = 0.46). There was a moderate correlation between advanced levels of cognitive impairment (MMSE scores 0–17) and institutionalisation at both 60 (*r* = 0.310, *p* = 0.003) and 90 days (*r* = 0.45, *p* < 0.001) of admission. The GDS and TUG were not correlated with achieving ‘stability’ or ‘decline’ over the duration of the admission.

#### 3.1.2. Associations 

After adjusting for age and sex, high ISAR scores (between 5–6 points) were significantly associated with clinical decline, i.e., remaining ‘unstable’ or ‘deteriorating’ at both 60 (OR 3.25, 95% CI: 1.84–5.74, *p* < 0.001) and 90 days (OR 3.07, 95% CI: 1.75–5.40, *p* < 0.001). Similarly, having high pressure ulcer risk (Walsall) scores was strongly associated with ‘decline’ at 60 (OR 4.92, 95% CI: 2.48–9.74, *p* < 0.001) and 90 days (OR 8.86, 95% CI: 3.48–22.5, *p* < 0.001). This was also observed for the MUST with high scores (≥2 suggestive of malnutrition) associated with remaining unstable at 60 days (OR 1.73, 95% CI: 1.01–2.98, *p* = 0.049) and 90 days (OR 2.33, 95% CI: 1.24–4.35, *p* = 0.008). Functional status (poor BI scores) was strongly associated with remaining ‘unstable’ or ‘deteriorating’ at both 60 (OR 6.41, 95% CI: 2.77–14.8, *p* < 0.001) and 90 days (OR 7.73, 95% CI: 3.20–18.6, *p* < 0.001). Finally, having more advanced cognitive impairment (MMSE scores of 0–17) at 60 days significantly determined remaining ‘unstable’ or ‘deteriorating’ (OR 2.08, 95% CI: 1.11–3.94, *p* = 0.02). This remained a strong association at 90 days (OR 4.23, 95% CI: 1.98–9.07, *p* < 0.001). 

The Rockwood CFS was not associated with ‘decline’ at 60 days but was significantly associated with this outcome at 90 days (OR 3.29, 95% CI: 1.55–6.99, *p* = 0.002). Prolonged TUG test times (>30 s) were not statistically significantly associated with ‘decline’ at either time point. Similarly, lower GDS scores (scores 0–5 indicating lower risk of depression) were not significantly associated with ‘decline’. 

### 3.2. Relationship between Instrument Scores on Admission and Risk of Institutionalisation, Hospitalisation and Death

#### 3.2.1. Institutionalisation

The adjusted associations between each risk score at baseline and the secondary outcomes at last follow-up are presented in Table 7 for the total sample (n = 88). There was a statistically significant risk of institutional care at follow-up among patients with higher Rockwood CFS scores (OR 2.29, 95% CI: 1.02–5.16, *p* = 0.045). Higher Walsall scores (>15 indicating high risk) were also associated with institutionalisation (OR 2.00, 95% CI: 1.2–3.33, *p* = 0.008). Institutionalisation at the end of the study period was also observed in those unable to complete the TUG or with prolonged times (>30 s) (OR 1.27, 95% CI: 1.03–1.57, *p* = 0.023). The majority of patients had severe levels of frailty measured on admission to the CVW. No other significant associations were found.

#### 3.2.2. Hospitalisation

Increased risk of hospitalisation was only statistically significantly associated with higher TUG scores (>30 s), (OR 1.29, 95% CI: 1.01–1.65, *p* = 0.039 *). 

#### 3.2.3. Death

Higher scores on the Rockwood CFS (indicating greater frailty severity; OR 2.80, 95% CI: 1.18–14.6, *p* = 0.049), higher MMSE scores (OR 3.16, 95% CI: 1.09–9.12, *p* = 0.034) and lower BI scores (indicating less independence; OR 2.75, 95% CI: 1.04–7.25, *p* = 0.041) were statistically significantly related with death. There were no other statistically significant findings.

## 4. Discussion 

This study found that several of the risk-screening instruments used in the North Dublin CVW were associated and correlated with pre-defined clinical health states (i.e., reaching a level of ‘stability’ versus remaining ‘unstable’ or ‘deteriorating’ examined as a composite outcome) in a sample of 88 frail older adults under active management and monitoring. After adjusting for age and sex, most measures were significantly associated with clinical ‘decline’ (composite outcome), while under follow-up, only frailty (at 60 days), the TUG and the GDS at both time points were not statistically significantly associated with clinical ‘decline’. Selected screening instruments measuring frailty (CFS), pressure ulcer risk (Walsall) and general mobility (TUG) at baseline were associated with increased likelihood of institutionalisation during the three months of follow-up. Baseline frailty (over two times), cognition (over three times) and functional ability (approximately three times) were associated with increased odds of death. Only prolonged TUG test scores indicating overall poor mobility and higher risk of falls were associated with hospitalisation, resulting in a 1.3 greater odds of at least one admission over follow-up. 

As the complexity of patients’ care needs is increasing in the community, it is important to be able to access and utilise a suite of screening tools for multidimensional assessment to quickly identify care needs, risk-stratify patients and understand the impact of the care delivered [2]. This study shows that a multidimensional approach to assessment in determining transitions in clinical status was required across a range of domains. This is important to assist in determining the likely outcomes of care and dependency levels to assist healthcare professionals in decision-making and impact of the model of care delivery [33]. In this study, moving from a pre-defined health state on admission (‘unstable’) to another (‘stable’ versus remaining ‘unstable’ or ‘deteriorating’) was important in establishing outcomes. These transitions were influenced by the level of functional dependence, mobility, nutritional status, pressure ulcer risk, cognitive levels, mood and risk of hospital admission (ISAR score) in a markedly frail older population. Depression (based on the GDS) scores were not. 

Determining outcomes in populations living with frailty can be challenging due to the complex interaction of multiple health factors [5]. Therefore, knowledge of the areas that may influence outcomes following interventions is important. This paper suggests that these can in part be guided by the results of a suite of screening tools. These transitions in pre-defined health states, whose likelihood is marked by scores in these instruments at 60 and 90 days, provides the basis for the development of a capability framework. This can be used as part of active monitoring of care to support ongoing assessments and directing interventions. Capability frameworks of care can have positive effects on the optimisation of care in areas such as chronic illness [34]. They can provide better approaches to assessment, care planning and team management [34]. The scores of screening tools have been associated with adverse outcomes in other settings [2,5,35,36,37,38].

In community-dwelling settings, the process of structured case-load management using an integrated approach to care is essential to improve healthcare outcomes for older populations living with frailty [39,40]. In this study, TUG scores at baseline were significantly associated with institutionalisation (1.27 times increased odds), indicating that these could be a useful guide. Participants were significantly more likely to be institutionalised if unable to complete the TUG or if durations exceeded 30 s, or if they had more advanced frailty or risk of pressure ulcers. Slower TUG times are useful in predicting a range of outcomes in older populations including falls, new impairment in ADLs and decline in overall health [35,41]. A decline in MMSE scores was associated with increased odds of clinical ‘decline’ and mortality (a three-fold increased odds). TUG scores have been linked to cognitive impairment and poorer performance outcomes overall and mortality among older community dwellers [35]. Those with more advanced levels of cognitive impairment (0–17) were significantly more likely to remain ‘unstable’ or ‘deteriorate’ at both 60 and 90 days (between two and four times the odds at 60 and 90 days, respectively). Risk of clinical ‘decline’ was also associated with the level of risk for re-admission (ISAR > 4) (three-fold increased odds) and higher risk of malnutrition (MUST ≥ 1) (approximately two-fold increased odds). Higher risk of malnutrition has been associated with greater levels of dependency, poor mobility levels and worsening cognition [35,41]. Hence, early identification and monitoring using a standardized nutritional screen to identify those at greatest risk is recommended [42,43]. 

Multi-factorial screening is important in older populations due to a higher level of co-morbidities, risk of frailty and musculoskeletal and sensory changes associated with ageing that increase the risk of adverse events and outcomes [33,44,45,46,47]. The core domains of screening as part of risk stratification in the CVW model translate into everyday clinical practice and provide a framework to develop pathways of care based on complex assessment, risk stratification and measurement of responses over time. This is particularly relevant with the development of new approaches to care delivery in the community such as the enhanced community care model in Ireland, which will enhance monitoring and access to specialist care in areas such as chronic disease management and older persons care [48]. One of the priorities of the enhanced community care model is to develop and implement alternative care pathways in the community and link with acute hospital services. Therefore, determining an escalation pathway for specialist input based on objective data through the use of easy-to-use, brief screening tools will be key to ensure timely access and reduce the risk of adverse outcomes [2]. 

This study found that individuals with high risk scores across several domains including cognition, nutrition, function and mobility were refractory to interventions and/or showed few signs of improvement. Frailty status is seen as an independent risk factor for higher morbidity and mortality rates and need for institutional care [21]. This may indicate that the care requirements of such individuals with advanced frailty have little scope for reversibility and that they may benefit less from admission to the nurse-led CVW. Instead, these may potentially benefit from additional expert specialist input [49]. Moreover, in this cohort, the risks of institutionalisation continued to remain high even with specialist input if they were severely frail, once other healthcare factors showed advancement towards higher levels of poor nutritional status, advancing cognition and lower levels of mobility, suggesting that additional supports including palliative care may be more beneficial to support these older patients and their families, irrespective of whether they transition to long-term care. Nevertheless, predicting those that are likely to respond to intensive specialist programmes of care to delay the need for institutional care can be challenging and highlights the demand for capability frameworks for care delivery. 

In an older population, measures of physical function such as lower levels of mobility and walking speed rather than diseases have been linked to poor outcomes such as death or institutionalisation [35,41,44,45,50,51]. In this study, participants with lower levels of mobility (assistance of one or two) were more likely to be ‘unstable’ or ‘deteriorating’, even after an intensive case management programme with specialist input. Future studies should examine frailty transitions in CVWs over time as such changes are associated with increased rates of mortality and are reportedly influenced by a range of both modifiable and non-modifiable variables including age, female gender, cardiovascular disease, cognitive impairment, obesity, sedentary lifestyles, smoking and social deprivation [51]. Determining appropriate timeframes for care delivery for frail older populations in the community is important when considering resources, quality of care and patient safety. In this study the 60-day time point after admission was the most significant time point in determining study outcomes. This time period was significant in measuring clinical health state as a predictor of outcome and also care domains such as cognition level, mobility, function and pressure ulcer risk to determine primary and secondary study outcomes. As such, it is important to adopt a multifactorial approach to examine characteristics as global measures of risk that are associated with frailty and assist in determining potential outcomes [44,45,46,47,51]. This includes realistic measures in the presence of frailty duration to transition from one clinical state to another. In this study, the 60-day time period was also a reasonable period for patients to transition from an ‘unstable’ to ‘stable’ clinical state and maintain this up to 90 days or to remain ‘unstable’ or ‘deteriorate’ following admission to the CVW. This is an important time point for frail older adults as the highest risk for adverse outcomes such as readmission occurs in the first 30 days after discharge to the community [19]. Furthermore, recovery periods for frail older persons almost double [20], and therefore, enhanced CVW monitoring and increased interventions during this time period could support older adults to remain at home. Timely access to increased supports and services delivered in discreet care bundles have shown potential to reduce unplanned hospital care and emergency department presentations in other settings [33,46,49], though further research using randomised controlled trials designs are needed to confirm this in a CVW model. 

Patients with greater levels of mobility and function, low to moderate frailty levels, low to moderate risk of cognitive impairment and lower risks of malnutrition or pressure ulcer development were more likely to be at home at the end of the study period. These results support previous research studies [31,33,35,36,37,38,44,45,46,47] and suggest that tighter inclusion criteria based on these brief risk-prediction tools may help identify those most likely to benefit from admission to the CVW. Given that this is a limited resource, an economic analysis of such models using different admission criteria and clear stratification of patients should be undertaken as a priority [2]. 

## 5. Limitations 

There are several limitations of the study. These include the lack of a comparator group to investigate potential differences in risk stratification and study outcomes for those admitted to the CVW compared with those receiving standard care within the community. This is a single-centre study in one country, limiting the generalisability of the findings. Further, based on the mean CFS score on admission, most patients had severe frailty, which also reduces the generalisability of the results to populations living with more advanced levels of frailty. Additional study with patients across the frailty spectrum would help better understand the utility of the screening instruments and the benefits of admission to the CVW model. As follow-up times were limited, further research should examine the long-term outcomes and their implications for the model. Longer term outcomes were not measured, potentially missing an additional effect. As such, risk screening and observations are specific to CVW admission. Finally, multiple screening instruments were assessed, increasing the risk of multiplicity and hence the possibility of type I errors.

## 6. Conclusions 

As the complexity of care needs among community-dwelling older adults increases, it is essential to prioritise those at risk of further deterioration and to identify those that are most likely to respond to and benefit from interventions. This study highlights the utility of several commonly used short risk screening tools employed as part of a CVW model of care to inform assessment of care needs and outcomes among an older and frail cohort. These can assist in developing a capability framework of care to inform decision-making and escalation pathways to specialist services as part of an enhanced approach to community care. Though further research is needed to identify which is the optimal suite of screening instruments in those with and without frailty and whether such risk stratification can improve outcomes for these patients, these results highlight the potential utility of such an approach.

The researchers would like to acknowledge and thank the Nursing and Midwifery Planning and Development Unit in North Dublin Ireland, the School of Nursing and Midwifery, Royal College of Surgeons Ireland and the Chief Nurse’s Office, Department of Health Ireland.

## Figures and Tables

**Figure 1 ijerph-18-05601-f001:**
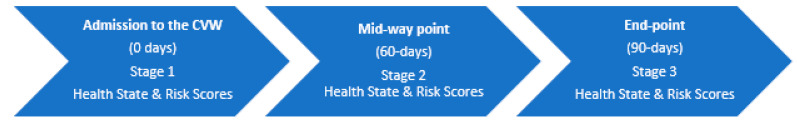
Application of a Markov model for a Community Virtual Ward (CVW) model study design.

**Table 1 ijerph-18-05601-t001:** Pre-defined ‘health states’ used to compare risk scores at 60 and 90 days with baseline, after admission to the Community Virtual Ward.

**Stable**	Ability to eat and drink returned (if previously diminished) Mental status considered normal or back to previous status if recently changed Functionally returning to their usual activities of daily living either independently or with support Improvements in emotional/psychological state or no evidence of deterioration No subsequent events in the last 30 days.
**Unstable**	Reduced or inadequate oral and/or nutritional intake Gradual cognitive decline or change in mental state Functionally unable to undertake their normal activities of daily living Social care needs exceeding supports within the home Secondary event resulting in above
**Deteriorating**	Increase in events and episodes (set of services provided to treat a clinical condition or procedure) Decrease in function and mobility (activities of daily living) Deterioration in mental status Further weight loss despite interventions

**Table 2 ijerph-18-05601-t002:** Admission criteria to the Community Virtual Ward (CVW).

Admission to CVW	CVW Selection
Diagnosis of frailty with evidence of at least one of the following: > Clinical deterioration > Increase in social care needs > Functional deterioration > Cognitive changes > Behavioural/emotional changes	Red CVW Event(s) occurred in the last 30 days or the patient was discharged from hospital in the last 30 days
	Amber CVW Event(s) occurred >30 days with evidence of more gradual decline in the last 3 months
	Green CVW Admission to the green VW can only occur following a period of monitoring either in the Red or Amber CVW. Admission to this ward is part of enhanced discharge planning including members of the primary care team and/or Specialist Geriatric Services.

**Table 3 ijerph-18-05601-t003:** Admission characteristics including demographics, co-morbidity and recent healthcare utilisation of patients (n = 88) in the Community Virtual Ward.

Variable	Number (%)/ Mean and Standard Deviation (SD)
Demographics Age (years)	82.83 (SD 6.406)
Sex	
Female	58 (65.9)
Male	30 (34.1)
Living Alone	
Yes	33 (37.5)
No	55 (62.5)
Co-morbidity Number of co-morbidities	2.82 (SD 1.034)
Number of medications	8.24 (SD 3.655)
Number of falls (last 3 months)	
No Falls	37 (42)
1 Fall	20 (22.7)
2 or More	31 (35.2)
Incontinence	
Yes	64 (72.7)
No	24 (27.3)
Unscheduled Healthcare Utilisation Unplanned admissions (3 months prior to CVW admission)	
1 hospital admission	36 (40.9)
2 or more hospital admissions	21 (23.9)
Emergency Department Presentations (last 3 months)	
1 ED presentation	36 (40.9)
2 or more ED presentations	31 (35.2)
Signs of Self-Neglect	
Yes	44 (51.1)
No	42 (48.9)

**Table 4 ijerph-18-05601-t004:** Risk-screening instruments used in the Community Virtual Ward model to predict outcomes and cut-off scores.

Risk-Screening Tool	Cut-Off Scores
Rockwood Clinical Frailty Scale	3
Timed up and Go Test	>13 s
Modified Functional Independence Measure	>1
Modified Barthel Index	16
Walsall Pressure Ulcer Risk Tool	>3
Malnutritional Universal Screening Tool	0
Mini Mental State Examination	25
Geriatric Depression Scale	4
Identification of Seniors at Risk tool	≥2

**Table 5 ijerph-18-05601-t005:** Correlation between risk-prediction scores and remaining ‘unstable’ or ‘deteriorating’ at 60 days and 90 days (n = 88).

Risk Scores	Correlation (*r*)	*p* Value
60 days		
Rockwood CFS	0.57	<0.001 ***
Walsall	0.59	<0.001 ***
Mobility (FIM)	0.56	<0.001 ***
MUST	0.22	0.039 *
TUG	0.15	0.154
ISAR	0.44	<0.001 ***
MMSE	0.26	0.015 *
Barthel	0.54	<0.001 ***
GDS	0.09	0.431
90 days		
Rockwood CFS	0.44	<0.001 ***
Walsall	0.68	<0.001 ***
Mobility (FIM)	0.58	<0.001 ***
MUST	0.32	0.002 **
TUG	0.09	0.393
ISAR	0.45	<0.001 ***
MMSE	0.46	<0.001 ***
Barthel	0.60	<0.001 ***
GDS	0.01	0.947

(* *p* < 0.05, ** *p* < 0.01, *** *p* < 0.001).

**Table 6 ijerph-18-05601-t006:** Association between risk-prediction scores and remaining ‘unstable’ or ‘deteriorating’ at 60 days and 90 days (n = 88), with adjusted (age and sex) odds ratio (OR) with 95% confidence intervals (CI).

Risk Scores	Odds Ratio	Lower 95% CI	Upper 95% CI	*p* Value
60 days				
Rockwood CFS	1.77	0.79	22	0.960
Walsall	4.92 ^	2.48	9.74	<0.001 ***
Mobility (FIM)	2.97 ^	1.81	4.86	<0.001 ***
MUST	1.73	1.01	2.98	0.049 *
TUG	1.09	0.74	1.62	0.669
ISAR	3.25 ^	1.84	5.74	<0.001 ***
MMSE	2.08	1.11	3.92	0.02 *
Barthel	6.41 ^	2.77	14.8	<0.001 ***
GDS	1.40	0.83	2.38	0.213
90 days				
Rockwood CFS	3.29 ^	1.55	6.99	0.002 **
Walsall	8.86 ^	3.48	22.5	<0.001 ***
Mobility (FIM)	3.08 ^	1.89	5.03	<0.001 ***
MUST	2.33 ^	1.24	4.35	0.008 **
TUG	1.03 ^	0.78	1.35	0.849
ISAR	3.07 ^	1.75	5.40	<0.001 ***
MMSE	4.23 ^	1.98	9.07	<0.001 ***
Barthel	7.73 ^	3.20	18.6	<0.001 ***
GDS	1.08 ^	0.62	1.91	0.778

(* *p* < 0.05, ** *p* < 0.01, *** *p* < 0.001, ^ intercept statistically significant at *p* < 0.05).

**Table 7 ijerph-18-05601-t007:** Adjusted odds ratio (OR) with 95% confidence intervals (CI) for risk of institutionalisation, hospitalisation and death associated with each baseline risk score; n = 88.

Baseline Risk Scores	OR	95 CI Lower	95 CI Upper	*p* Value
Institutionalisation				
Rockwood CFS	2.29	1.02	5.16	0.045 *
Walsall	2.00	1.20	3.33	0.008 **
Mobility (FIM)	1.19	0.98	1.43	0.080
MUST	0.84	0.48	1.46	0.530
TUG	1.27	1.03	1.57	0.023 *
ISAR	1.47	0.91	2.37	0.118
MMSE	1.12	0.59	2.14	0.722
Barthel	1.70	0.88	3.26	0.114
GDS	1.20	0.70	2.06	0.515
Hospitalisation				
Rockwood CFS	1.30 ^	0.56	3.01	0.542
Walsall	0.79	0.48	1.29	0.347
Mobility (FIM)	0.98	0.79	1.23	0.890
MUST	0.77	0.40	1.50	0.440
TUG	1.29	1.01	1.65	0.039 *
ISAR	1.55	0.87	2.77	0.137
MMSE	1.13	0.53	2.41	0.749
Barthel	0.73	0.35	1.54	0.407
GDS	1.42	0.75	2.68	0.283
Death				
Rockwood CFS	2.80	1.18	8.23	0.049 *
Walsall	1.69	0.83	3.46	0.150
Mobility (FIM)	1.12	0.87	1.43	0.390
MUST	0.83	0.37	1.86	0.651
TUG	0.92	0.69	1.23	0.561
ISAR	1.69	0.84	3.43	0.144
MMSE	3.16	1.09	9.12	0.034 *
Barthel	2.75	1.04	7.25	0.041 *
GDS	1.10	0.53	2.29	0.800

(* *p* < 0.05, ** *p* < 0.01, ^ intercept statistically significant at *p* < 0.05).

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
