# Peer review of "Utilisation of a Suite of Screening Tools to Determine Adverse Healthcare Outcomes in an Older Frail Population Admitted to a Community Virtual Ward"

_ijerph, 2021, doi:10.3390/ijerph18115601_

Round 1

Reviewer 1 Report

I think this article is interesting, well-written, and contains important information about frailty of community-dwelling older patients in a Community Virtual Ward. This work yields insight into risk and patient outcomes for this group and will help with decision-making as it relates to these patients. While I like the sample selected, research design, and instruments used, I have concerns about the statistics, which I have listed out below.

Abstract

Page 1, line 22: please spell out MMSE prior to using it as an abbreviation

Materials and Methods

  1. Page 5, last paragraph: Why did you run a series of ANOVAs to test significance of a Spearman’s correlation when the Stata procedure produces significance? Or did I misunderstand your justification and the ANOVAs ran for another reason? If so, please state clearly in the text as the ANOVAs are not necessary without further justification. Also, post-hoc tests should be ran on the ANOVA results if interested in differences between clinical health states to identify which state(s) were different from the other(s).
  2. I would highly suggest running a single ordinal logit or probit model with clinical health state as the outcome variable and the risk scores as independent variables. Then, the risk scores can be evaluated while taking into account each other and any confounders can be identified. Odds ratios are provided, which give a measure of the odds of these tools’ relationship with clinical states.

Results

  1. Section 3.1, 3.2, & 3.3: the reader would greatly benefit from a table summarizing this information. Also, it will lead to a handy tool for practitioners where they can look up your results and pick the instrument that would best evaluate their patient’s outcome.
  2. Section 3.1, 3.2, & 3.3: what test(s) are these results from? If these are ANOVAs, what is the F-value? How about df?

Discussion

Which risk tool was most/least/not important would aid the Discussion, which would come from running a logit or similar statistical model. Also, I would like to know which clinical health state(s) were different than the other(s). Those are two really important pieces of information to be able to describe risk for these clinical states.

Reviewer 2 Report

This is a valuable contribution to knowledge, but some changes need to be made.

  1. The abstract should be rewritten. As it is now, the reader does not get a clear idea of the results and their usefulness. In addition, it has been detected the use of some acronyms that are either incorrect or have not been previously defined.
  2. It would be necessary to improve the introduction, as there are parts of it that are not appropriate for this point in an article. In addition, more evidence should be provided to determine the importance of the research. For example, lines 49 to 59 are more appropriate for the methodology section than for an introduction.
  3. It is necessary to provide more information about the selected sample.
  4. Perhaps it would be appropriate to introduce some table or figure that will facilitate the follow-up and reading of the results found in the article.
  5. Include n (sample size) in the text for each cluster or group.

The article is well-written, compact and specific. The research description is clear and structured. Well done.

Round 2

Reviewer 1 Report

Thank you for your changes. Wow! The logit has greatly improved your paper and your findings. I really like how this turned out. There are still a few areas of refinement that come from the major overhaul of the statistics you made. They are clarifications, primarily, and should be quick to address. They will make your paper strong for application by clinicians and public health experts and understandable to the reader.

Materials and Methods

Pg 7.

Were your data normally distributed, allowing for a Pearson’s correlation? Also, why do a correlation when you are doing a logistic regression, which identifies any associations? I think the correlations take away from your logit, which is extremely powerful and gives great information, including your OR. Your OR are so informative!

BMI was added as a predictor, but BMI is racially and gender biased. Why use this measure? BMI is not actually the cause of anything or indicative of anything for these patients, right? But, allostatic load can inform about a patient’s overall health, particularly in your sample of older and frail individuals. So, please remove BMI and substitute it with measure(s) of allostatic load (or drop that information all together and state it as a bias, or use waist circumference).

The approach with logistic regression is much improved. But, what type of logistic regression did you run? That is unclear from the text. Also, even though “health states” is listed at the outcome variable, what does this actually mean? Clarification in text as to what the outcome variable was will help to evaluate what was ran and the meaning behind your results.

The combined correlation and logit table is hard to interpret. What are the intercepts/constants (I don’t see them). I also don’t understand how you have separate coefficients for 60 and 90 days, but this will probably be understood once the previous comments have been clarified.

Pg 9., line 315 – the OR is missing in the middle of that line

Results

Why are the OR not interpreted? You have some ORs that are quite large and indicate a huge increase of risk. Interpretation of what these mean (e.g., 5-fold higher odds) will greatly improve the impact of your paper and allow clinicians to understand how much risk is associated with each item.

Author Response

Thank you for your changes. Wow! The logit has greatly improved your paper and your findings. I really like how this turned out. There are still a few areas of refinement that come from the major overhaul of the statistics you made. They are clarifications, primarily, and should be quick to address. They will make your paper strong for application by clinicians and public health experts and understandable to the reader.

Response: The authors agree and thank the reviewer for their considered and constructive review. New edits and additional analysis have been highlighted in the main text for clarity.

Materials and Methods

Pg 7. Were your data normally distributed, allowing for a Pearson’s correlation?

Response: We have confirmed that the majority of the data were non-normally distributed, likely due to the small and homogenous sample. We have changed the correlational analysis to Spearman’s rho, accordingly.

Also, why do a correlation when you are doing a logistic regression, which identifies any associations? i think the correlations take away from your logit, which is extremely powerful and gives great information, including your OR. Your OR are so informative!

Response: Correlations were included as they were initially used to help predict likely associations (to quickly identify the strength and direction between the variables) before the regression modelling. While we have updated these correlations as described above and included them – they can be removed if required.  We have included these in a new separate table.

BMI was added as a predictor, but BMI is racially and gender biased. Why use this measure? BMI is not actually the cause of anything or indicative of anything for these patients, right? But, allostatic load can inform about a patient’s overall health, particularly in your sample of older and frail individuals. So, please remove BMI and substitute it with measure(s) of allostatic load (or drop that information all together and state it as a bias, or use waist circumference).

Response: The authors agree and have now removed BMI and re-run the analysis accordingly.

The approach with logistic regression is much improved. But, what type of logistic regression did you run? That is unclear from the text. Also, even though “health states” is listed at the outcome variable, what does this actually mean? Clarification in text as to what the outcome variable was will help to evaluate what was ran and the meaning behind your results. The combined correlation and logit table is hard to interpret. What are the intercepts/constants (I don’t see them). I also don’t understand how you have separate coefficients for 60 and 90 days, but this will probably be understood once the previous comments have been clarified.

Response: The authors created a generalised linear model, adjusted in the updated manuscript for age and gender. This has been made clear in the methods. This study developed three ‘health states’, which were defined a priori as part of the development of the CVW model. These were defined for each patient at each time point by the CVW manager (stable, deteriorating and unstable). These are provided in detail in Table 1. To facilitate analysis these were dichotimised into a composite outcome of remaining ‘unstable’ or ‘deteriorating’ called ‘clinical decline’ (1) versus ‘stable’ (0). These were measured at baseline (entry into the study) and again at 60 days and finally 90 days.

The criteria to define these ‘health states’ has been clarified in the text and is presented in precise detail in Table 1 (page 2).

Intercepts (constants) were available (part of each model) and calculated as part the regression models – The OR for the intercept was zero or close to this for all models run. Those that were statistically have now been identified in each table (^).

Pg 9., line 315 – the OR is missing in the middle of that line

Response: This has been corrected.

Results

Why are the OR not interpreted? You have some ORs that are quite large and indicate a huge increase of risk. Interpretation of what these mean (e.g., 5-fold higher odds) will greatly improve the impact of your paper and allow clinicians to understand how much risk is associated with each item.

Response: The authors agree and have now included these in the discussion.